# Effect of TiO_2_ Additives on the Stabilization of *h*-YbFeO_3_ and Promotion of Photo-Fenton Activity of *o*-YbFeO_3_/*h*-YbFeO_3_/*r*-TiO_2_ Nanocomposites

**DOI:** 10.3390/ma15228273

**Published:** 2022-11-21

**Authors:** Sofia Tikhanova, Anna Seroglazova, Maria Chebanenko, Vladimir Nevedomskiy, Vadim Popkov

**Affiliations:** 1Ioffe Institute, 194021 Saint Petersburg, Russia; 2Saint-Petersburg State Institute of Technology, 190013 Saint Petersburg, Russia

**Keywords:** ytterbium orthoferrite, titanium dioxide, solution combustion synthesis, heterojunctions, photocatalysts, Fenton-like processes

## Abstract

Nanostructured hexagonal rare-earth orthoferrites (*h*-RfeO_3_, R = Sc, Y, Tb-Lu) are well known as a highly effective base for visible-light-driven heterojunction photocatalysts. However, their application is limited by metastability, leading to difficulties in synthesis due to the irreversible transformation to a stable orthorhombic structure. In this work, we report on a simple route to the stabilization of *h*-YbFeO_3_ nanocrystals by the synthesis of multiphase nanocomposites with titania additives. The new I-type heterojunction nanocomposites of *o*-YbFeO_3_/*h*-YbFeO_3_/*r*-TiO_2_ were obtained by the glycine–nitrate solution combustion method with subsequent heat treatment of the products. An increase in the mole fraction of the *h*-YbFeO_3_ phase in nanocomposites was found with the titanium addition, indicating its stabilizing effect via limiting mass transfer over heat treatment. The complex physicochemical analysis shows multiple contacts of individual nanocrystals of *o*-YbFeO_3_ (44.4–50.6 nm), *h*-YbFeO_3_ (7.5–17.6 nm), and rutile *r*-TiO_2_ (~5 nm), confirming the presence of the heterojunction structure in the obtained nanocomposite. The photocatalytic activity of *h*-YbFeO_3_/*o*-YbFeO_3_/*r*-TiO_2_ nanocomposites was evaluated by the photo-Fenton degradation of the methyl violet under visible light (λ ≥ 400 nm). It was demonstrated that the addition of 5 mol.% of TiO_2_ stabilizes *h*-YbFeO_3_, which allowed us to achieve a 41.5 mol% fraction, followed by a three-time increase in the photodecomposition rate constant up to 0.0160 min^−1^.

## 1. Introduction

Metal oxide nanoparticles have been well-described in a number of applications involving biomedicine and industry [1,2,3]. As a perspective class of oxide-based materials, orthoferrites of rare earth elements have gained considerable interest among researchers due to their structural [4,5,6,7,8], magnetic [6,9,10,11], electrical [6,12], and photocatalytic [13,14,15,16,17,18] properties. This class of compounds has found practical applications as a material for catalysts [19], magnetic resonance imaging [20,21], bio-medical drug delivery [22,23], sensors [24,25], and pigments [26,27]. Notably, ytterbium orthoferrite can be used as a highly efficient photocatalyst whose activity depends on its crystal structure [28]. YbFeO_3_ is characterized by an orthorhombic perovskite structure (*Pbnm/Pnma*), which is thermodynamically stable [29]. However, a hexagonal phase was also obtained, first in thin films and then in bulk form. Depending on the synthesis conditions, the hexagonal phase can be either polar P63 cm or nonpolar P63/mmc [30]. It is known that substances in the metastable states often exhibit unique properties, for example, higher specific surface area [31] and narrower distribution of crystallite sizes [13].

On the one hand, it has been reported that the hexagonal form (*h*-YbFeO_3_) demonstrates higher photocatalytic activity than that of *o*-YbFeO_3_ under the same conditions [28]. On the other hand, *h*-YbFeO_3_ shows high instability and a tendency to transform easily in the stable *o*-YbFeO_3_ phase [32,33].

Previously, several ways of stabilization by creating spatial constraints were found. Some of the most common are self-organized [34,35] and template-assisted combustion processes [36]. However, the stabilization effect of the second phase addition has never been reported. Recently, we confirmed the possibility of the stabilization of *h*-YbFeO_3_ by the CeO_2_ extra phase [32]. In the present work, for the first time, the possibility of creating additional spatial restrictions due to the insertion of the second TiO_2_ phase was demonstrated and the synthesis of novel I-type heterojunction nanocomposites of *o*-YbFeO_3_/*h*-YbFeO_3_/*r*-TiO_2_ was performed.

In this work, TiO_2_ was used as an extra phase because of its high photocatalytic activity as a co-catalyst in the photo-Fenton-like degradation processes [37]. Additionally, titania dioxide is a highly stable and environmentally safe photocatalyst in the near-UV or even visible light (due to dye-driven photo-sensibilization) regions [38,39]. To promote the uniform distribution of the compounds, the solution combustion method was used and successfully executed.

## 2. Materials and Methods

### 2.1. Synthesis of YbFeO_3_-Based Photocatalysts

#### 2.1.1. Synthesis of the Titanyl Nitrate TiO(NO_3_)_2_ Solution

The synthesis of *h*-YbFeO_3_/*o*-YbFeO_3_/TiO_2_ nanocomposites included the preparation of titanyl nitrate from titanium tetrachloride TiCl_4_. Firstly, distilled water was added to 1 mL of TiCl_4_ with constant stirring to form a clear solution of TiCl_4_. Then, 2 mL of 5% solution of NH_4_OH was added to obtain a white precipitate of TiO(OH)_2_. Finally, it was completely dissolved in concentrated HNO_3_. The process can be described by the following reactions:TiCl_4_ + 3 NH_4_OH = TiO(OH)_2_ + 3 NH_3_ + 4 HCl(1)
TiO(OH)_2_ + 2 HNO_3_ = TiO(NO_3_)_2_ + 2 H_2_O(2)

#### 2.1.2. Synthesis of o-YbFeO_3_/h-YbFeO_3_/TiO_2_ Nanocomposites

Series of *h*-YbFeO_3_/*o*-YbFeO_3_/TiO_2_ samples with a molar fraction of titanium dioxide of 0, 2.5, 5, 7.5 mol.% were obtained by glycine–nitrate solution combustion method followed by subsequent heat treatment of the products. The precursor salts Yb(NO_3_)_3_·6H_2_O (99.5%) 0.5469 g, Fe(NO_3_)_3_·9H_2_O (98.0%) 0.5149 g, and glycine (C_2_H_5_NO_2_, 98.5%) 0.1168 g were solved in 10 mL of distilled water with 0.1, 0.2, 0.3, and 0.4 mL of titanyl nitrate TiO(NO_3_)_2_ solution (Section 2.1.1) for the 2.5, 5, and 7.5 samples, respectively. The considered combustion process is described by the following reaction scheme:9 Yb(NO_3_)_3_ · 5H_2_O + 9 Fe(NO_3_)_3_ · 9H_2_O + 9 TiO(NO_3_)_2_ + 40 C_2_H_5_NO_2_ = 9 (YbFeO_3_)(TiO_2_) + 56 N_2_ + 80 CO_2_ + 226 H_2_O(3)

To achieve the complete dissolution of components, the reaction solution was mixed for 15 min at room temperature. After that, the reaction mixture was heated in a sand bath until the water evaporated and the solution self-ignited. As a final product of combustion, the foam-like porous substance was pounded in a mortar to form a powder. Then it was calcinated at 800 °C for 24 h to achieve the complete removal of organic residues and form crystalline products. The ratio of the contents of glycine and nitrate ions (glycine to nitrate ratio or *G/N*) in the reaction solution equaled 0.2. In the excess of oxidizer, the glowing mode of combustion is applied, and the formation of X-ray amorphous products is promoted.

### 2.2. Characterization

The elemental composition was investigated by energy-dispersive X-ray spectroscopy (EDXS, Tescan Vega 3 SBH, Oxford INCA X-act, TESCAN Brno s.r.o., Brno, Czechia). The crystal structure was investigated by X-ray diffraction (XRD, Rigaku SmartLab 3, Rigaku Corporation, Tokyo, Japan) using Cu Kα radiation (λ = 1.540598 Å). The average crystallite size was calculated from the broadening of X-ray diffraction lines using the method of fundamental parameters implemented in the SmartLab Studio II software package (Rigaku Corporation, Tokyo, Japan). Quantitative X-ray phase analysis was conducted by means of the Rietveld method. The morphology and microstructure were investigated by scanning electron microscopy (SEM, Tescan Vega 3 SBH, TESCAN Brno s.r.o., Brno, Czechia)) and transmission electron microscopy (TEM, ZEISS Ultra 55 microscope, FELMI ZFE, Oberkochen, Germany) in the image and selected area electron diffraction (SAED) mode. The diffuse reflectance spectra in the UV–visible region (DRS, Avaspec-ULS2048, AvaSphere-30-Refl, Avantes, Apeldoorn, The Netherlands) were recorded at room temperature in the range of 400–750 nm using an AvaSphere-30-Refl integrating sphere. The specific surface of the samples, pore volume, and pore size distribution were determined by adsorption-structural analysis (ASA, Micromeritics ASAP 2020, Norcross, GA, USA). Sorption–desorption isotherms were obtained at a liquid nitrogen temperature of 77 K.

### 2.3. Photocatalytic Experiments

The photo-Fenton-like degradation of methyl violet (MV) with the addition of hydrogen peroxide was used to investigate the photocatalytic activity of the produced catalysts. The photocatalytic experiment was carried out under visible light irrigation λ_max_ = 410 nm. The typical process was described in detail in previous work [16]. To establish the adsorption equilibrium, the reaction solutions were stirred in darkness for 30 min before the start of the photocatalytic experiment. For the absorption spectra measurements and the concentration of the MV determination, AvaLight-XE light source and AvaSpec-ULS2048 spectrometer were used. Decolorization of the dye was measured every 10 min.

## 3. Results and Discussion

### 3.1. SEM, EDXS, and Elemental Mapping

The energy-dispersive X-ray spectroscopy (EDXS) results (Figure 1) demonstrate that, for all samples, the Yb: Fe ratio is approximately equal to 1. In the samples containing titanium dioxide, its amount corresponds to the values declared during the synthesis. Thus, the planned ratios of key elements are achieved within the error of the determination.

Energy dispersive elemental mapping of a sample containing 5 mol.% of TiO_2_ Figure 2c shows that the key elements are distributed uniformly. Single-element mapping Figure 2d,f confirms the absence of accumulation zones of individual chemical elements in the sample. Microphotography of the area used for mapping Figure 2a,b reflects the foam-like morphology typical for samples synthesized by the solution combustion method [40,41,42].

### 3.2. PXRD

Figure 3a demonstrates the PXRD patterns measured at room temperature with ground samples after heat-treatment at 800 °C. All peaks of YbFeO_3_ can be well-indexed to the characteristic XRD peaks of the orthorhombic structure according to the ICSD card No. 189735. The absence of any additional peaks indicates the successful synthesis of pure *o*-YbFeO_3_ by the solution combustion method. For all the samples synthesized with the addition of TiO_2_, the presence of *h*-YbFeO_3_ is observed. For samples containing from 2.5 to 7.5 mol.% of TiO_2,_ only *o*-YbFeO_3_, *h*-YbFeO_3_, and *r*-TiO_2_ phases can be indicated. There were no other peaks related to impurities for these samples. A further increase in the mole fraction of titanium dioxide leads to the formation of a mixture of phases *o*-YbFeO_3_, *h*-YbFeO_3_, Yb_2_Ti_2_O_7_, and Fe_2_O_3_.

Crystallite sizes for the major phases Figure 3b varied in the range of 44.4–50.6 nm for *o*-YbFeO_3_ and 7.5–17.6 nm for *h*-YbFeO_3_, which is in consonance with existing research. The minimum size of a *h*-YbFeO_3_ crystallite is observed for the sample containing 5 mol.% of TiO_2_. The determination of crystallite sizes for the minor phases is complicated due to the overlaying of the peaks.

The degree of conversion dependences in Figure 3c were evaluated by the Rietveld refinement method. Structural parameters and criteria of fit of the typical refinement for the sample containing 5 mol.% of TiO_2_ are presented in Figure 4. The refinement data confirms that the *o*-YbFeO_3_/*h*-YbFeO_3_/*r*-TiO_2_–5% sample consist of three phases: *o*-YbFeO_3_ with orthorhombic structure, space group *Pbnm*; *h*-YbFeO_3_ with hexagonal structure, space group *P63/mmc*; *r*-TiO_2_ with rutile structure, and space group *P 42/m n m.* The investigated unit cell parameters are in good agreement with literature data. The value of R*_wp_*, R*_p_*, and R*_e_* parameters show that the refinements were correct.

According to PXRD data, the dependences of the degree of conversion of the main crystalline phases on the proportion of added titanium dioxide were calculated Figure 3c. As the content of titanium dioxide in the composite increases, the mole fraction of TiO_2_ increases linearly, which is consistent with the EDX data. In addition, when titanium dioxide is added, the formation of the h-YbFeO_3_ is observed. The maximum molar fraction of h-YbFeO_3_ phase is 39 mol.% for the sample with 5 mol.% of TiO_2_. For the sample containing 10 mol.% of TiO_2_ the molar fraction of phase containing titanium agrees with the increase in the added amount of titanyl nitrate.

### 3.3. HR-TEM and SAED

The morphology and microstructure of the *h*-YbFeO_3_/*o*-YbFeO_3_/TiO_2_–5% sample were investigated by transmission electron microscopy (TEM) Figure 5. The TEM image of the nanocomposite presents the shaped crystals surrounded by a foam-like substance. As was determined by XRD, the crystalline size for *o*-YbFeO_3_ is approximately 50 nm, which means that we can assume that the shaped crystals are associated with *o*-YbFeO_3_, whose morphology is typical for heat-treated glycine-nitrate combustion products. The foam-like structure is more usual for products without heat treatment and for incompletely formed particles [28].

Therefore, the SAED pattern Figure 5d from Figure 5a confirms the absence of an amorphous phase. The dark-field TEM of single peaks Figure 5e, which are typical for monocrystals, confirms that the shaped crystals are *o*-YbFeO_3_. The smaller light areas in the foam-like zone show the mixture of phases in it. The dark-field TEM of multiple ring peaks Figure 5f, which are typical for polycrystals, show the presence of small crystals in interporous substances.

For clearer phase determination, d-spacing was calculated from the SAED pattern. According to the results presented in Table 1, rings 1, 4, 5 are associated with *o*-YbFeO_3_; the 2 is *r*-TiO_2_, and the 3 is *h*-YbFeO_3_. The crystallite size of *r*-TiO_2_ is about 5 nm due to TEM results. The colored scheme of relative phase distribution shows the individual phases are in multiple contacts, which confirms the formation of the nanocomposite with a heterojunction structure.

The formation of heterojunction was additionally confirmed by HR-TEM Figure 5c. The *d*-spacing was calculated from the HR-TEM image for the phase determination. As shown, the *o*-YbFeO_3_ crystal is surrounded by *r*-TiO_2_ and *h*-YbFeO_3_ crystals.

As was previously reported, the presence of an inert phase prevents mass transfer and interrupts the structural transition hexagonal → orthorhombic ytterbium orthoferrite [32]. It can be surmised that the formation of *o*-YbFeO_3_ was interrupted by *r*-TiO_2_ particles on the board between shaped and foam-like areas. The mass transfer from *h*-YbFeO_3_ to *o*-YbFeO_3_ becomes difficult and the formation of *h*-YbFeO_3_ crystals takes place. Consequently, we can assume that TiO_2_ has a stabilizing effect on *h*-YbFeO_3_.

### 3.4. Low-Temperature N_2_ Sorption–Desorption

For a more detailed study of the specific surface area, average pore size, and volume of the obtained nanopowders, an adsorption structural analysis was performed using a low-temperature adsorption–desorption of nitrogen (77 K). Figure 6 presents the results of the experiment. As can be noticed from Figure 6a, the N_2_ adsorption–desorption isotherm obtained for YbFeO_3_ presented the type II isotherm with H3 hysteresis loop, and the classical type IV isotherm with H4 hysteresis loop was obtained for other samples containing TiO_2_. The change in the isotherm type and the form of the loop results from the appearance of h-YbFeO_3_ in the samples [28].

The surface area of the *o*-YbFeO_3_/*h*-YbFeO_3_/TiO_2_ samples calculated by the BET method is presented as an inset in Figure 6a. According to the obtained data, the addition of up to 2.5 mol.% of titanium dioxide promotes an increase in the specific surface area from 9.1 to 13.0 m^2^/g. However, with further increase in the proportion of TiO_2_, a smoother asymptotic increase is observed, and for samples containing 5–10 mol.% of titanium dioxide, the value of the specific surface remains practically unchanged. The pure YbFeO_3_ sample presents a mesoporous structure, whereas other samples have a macroporous structure. The absence of micropores explains the relatively low surface area. Because of their small size, the *h*-YbFeO_3_ and TiO_2_ nanoparticles can fill the mesopores in the *o*-YbFeO_3_ matrix and decrease the average surface.

### 3.5. DRS

The optical absorption property of h-YbFeO_3_/o-YbFeO_3_/TiO_2_ was analyzed via UV-vis diffuse reflectance spectroscopy, and the results are shown in Figure 7a. The UV–visible absorption spectra of the synthesized nanocomposites were recorded in the range of wavelength of 400–750 nm. The light reflection spectra exhibit the reflection bands in the visible region, implying that those samples should be effective as visible light photocatalysts. Samples with 2.5 and 5 mol.% of TiO_2_ show the most intensive absorption in the visible light region, which makes them perspective for visible-light-driven photocatalytic processes.

The obtained samples demonstrate active absorption in the visible region of the spectrum, and the inflections in the reflection curves indicate the multiphase nature of the samples. The Tauc plot obtained by analyzing the reflectance measurement with Kubelka–Munk (KM) emission function to determine the band gaps of these phases Figure 7b, and the calculation results are presented in the inset. Considering that the calculated band gap values (E_g_) for each of the phases change unsystematically, it can be assumed that they do not depend on the proportion of added titanium dioxide. The average E_g_ values were 2.55 eV × (*o*-YbFeO_3_), 2.20 eV (*h*-YbFeO_3_), and 3.06 eV (TiO_2_), which corresponds with the existing research [43,44]. According to the literature data, the band gap value of 3.1 eV is associated with the rutile TiO_2_ [45]. It can be established that the resulting composite is based on individual phases of *o*-YbFeO_3_, *h*-YbFeO_3_, and *r*-TiO_2_.

As a result of calculations presented in Table 2, the electronic structure parameters of *o*-YbFeO_3_/*h*-YbFeO_3_/TiO_2_ nanocomposite were determined [28,46]. The relative position of the valence and conduction bands confirms the formation of a type I heterojunction structure, which leads to the suppression of recombination of electron–hole pairs in photo-Fenton-like processes.

### 3.6. Photo-Fenton-like Photocatalytic Activity

The photocatalytic activity of *o*-YbFeO_3_/*h*-YbFeO_3_/TiO_2_ nanocomposites was evaluated by the photodegradation of the typical organic dye methyl violet (MV) under visible light irrigation, and the time-dependent photodegradation of MV is illustrated in Figure 8a. The high rate of discoloration of the solution shows that the resulting nanocomposites are effective catalysts for Fenton processes [46].

The kinetic curves of the photodegradation of MV are presented in Figure 8b. To establish the adsorption–desorption equilibrium, the reaction solution in it was preliminarily kept in the dark for half an hour with continuous stirring. The kinetic curves showed that the degradation efficiency for each of the samples is different and depends on the mole fraction of TiO_2_. The sample with a TiO_2_ content of 5.0% demonstrated the highest photodecomposition efficiency, which amounted to 30.1%.

All kinetic dependences rearranged in ln(C/C_0_) = f(t) coordinates Figure 8c are linear with good R values, which confirms the pseudo-first-order of the reaction of methyl violet photodegradation that is typical for Fenton-like processes [47]. The rate constants of the photodecomposition were established as the tangent of the slope of the linearized kinetic curves Figure 8d.

At first, the values of the rate constants increase with the addition of up to 5% of TiO_2_ and decrease with the further addition of titanium dioxide. The initial increase is associated with an increase in the specific surface area and the proportion of the hexagonal phase, leading to an increase in charge separation/transfer. The subsequent decrease is due to the fact that with an increase in the added TiO_2_, the active centers of the photocatalyst are screened, which prevents the absorption of light, contributing to an increase in charge recombination.

Previously, it was shown that the addition of CeO_2_ to the YbFeO_3_-based nanocomposite positively affects the efficiency of photodegradation. The increase in photocatalytic activity is connected with the increment in the *h*-YbFeO_3_ molar fraction. In previous works [28,32], the growth of the *h*-YbFeO_3_ fraction in YbFeO_3_ from 10.2% to 18.8% led to an increase in the photodecomposition rate constant from 0.0048 min^−1^ to 0.0138 min^−1^. In the work, the addition of TiO_2_ stabilized *h*-YbFeO_3,_ and the fraction of 41.5% was achieved, which was followed by an increase in the photodecomposition rate constant to 0.0160 min^−1^.

### 3.7. Mechanism of Photo-Fenton-like Activity over o-YbFeO_3_/h-YbFeO_3_/TiO_2_ Nanocomposites

The principally possible mechanism of photo-Fenton-like degradation of MV by heterojunction nanocomposite *h*-YbFeO_3_/*o*-YbFeO_3_/TiO_2_ was presented in Figure 9 [37,47,48,49].

As the bang gap values E_g_ for *o*-YbFeO_3_, *h*-YbFeO_3_, and TiO_2_ (2.55, 2.2 and 3.06 eV, respectively), correspond to absorption in the visible region, the electron–hole pairs can be formed in the components of the nanocomposite under visible light irradiation. Since the TiO_2_ has the widest bang gap with the highest conduction band and the lowest valence band values among other components, the obtained nanocomposite is a combination of nested type I heterojunctions (straddling gap). Thus, the photo-generated electrons can transfer from TiO_2_ to *o*-YbFeO_3_ and then to *h*-YbFeO_3_.

After the electron–hole pair is generated on the TiO_2_ under visible light, the part of the photo-generated electrons (e^−^) from CB TiO_2_ reduces the surface cations of titanium (IV) (Ti^4+^) to titanium (III) cations (Ti^3+^); then hydrogen peroxide (H_2_O_2_) oxidizes the surface cations of titanium (III) (Ti^3+^) to titanium (IV) (Ti^4+^) to form hydroxyl radical (·OH) [37,48]. The reaction of ferrous ions generates highly reactive hydroxyl radicals capable of attacking and mineralizing organic pollutants. The other electrons from CB TiO_2_ can migrate to CB *o*-YbFeO_3_.

On the surface of *o*-YbFeO_3_, the generation of electron–hole pairs is also available. Part of the photo-generated electrons react with oxygen (O_2_) adsorbed on the surface of *o*-YbFeO_3_ to form a superoxide anion (·O_2−_); then the superoxide anion (·O_2_−) reacts with water (H_2_O) to form hydroxyl radical (·OH). The other electrons from CB *o*-YbFeO_3_ can migrate to *h*-YbFeO_3_, where they take part in the same processes.

On both the *o*-YbFeO_3_ and *h*-YbFeO_3_ phases the photo-generated electrons (e^−^) reduce the surface ferric cations (III) (Fe^3+^) to ferrous cations (Fe^2+^); then hydrogen peroxide (H_2_O_2_) oxidizes the surface ferrous cations (Fe^2+^) to ferric cations (III) (Fe^3+^) to form hydroxyl radical (·OH).

In turn, the photo-generated holes (*h^+^*) from VB TiO_2_ can react with water molecules (H_2_O) to form hydroxyl radicals (·OH) or migrate to the VB *o*-YbFeO_3_ and, finally, to *h*-YbFeO_3_ to participate in the formation of hydroxyl radicals (·OH) in the same process. Methyl violet is oxidized by ·OH to produce the two most stable products, CO_2_ and H_2_O.

It can be noticed that most of the generated electrons and holes migrate to the *h*-YbFeO_3_ cascade. This demonstrates the leading role of the hexagonal phase in the photo-Fenton-like photodegradation of the dye.

## 4. Conclusions

Thus, novel I-type heterojunction nanocomposites that are based on hexagonal and orthorhombic forms of YbFeO_3_ and titanium dioxide were successfully developed. Nanocomposites were obtained by the method of glycine–nitrate solution combustion with subsequent heat treatment of the products.

The formation features and the relative influence of the components of the nanocomposite on the formation and evolution of the polycrystalline system were studied in detail by the SEM, EDXS, and PXRD methods. The developed foamy morphology and highly porous structure cause high values of the specific surface area, contributing to an increase in the photocatalytic activity of the samples. The formation of an I-type heterojunction, confirmed by TEM and DRS data, promotes the suppression of the recombination of electron–hole pairs and positively affects the photocatalytic activity during the photo-Fenton-like photodecomposition of MV. Moreover, the addition of TiO_2_ extra phase contributes to the stabilization of the hexagonal phase. The optimal amount of added TiO_2_ was 5%, which is due to the high values of the specific surface area and the proportion of the hexagonal phase, as well as the processes of overlapping of the active centers of the catalyst surface by TiO_2_ particles and an increase in charge recombination. The resulting photocatalysts based on a heterojunction nanocomposite can be used as promising photocatalysts for processes such as photo-Fenton-like oxidation of organic pollutants and other advanced oxidation techniques.

## Figures and Tables

**Figure 1 materials-15-08273-f001:**
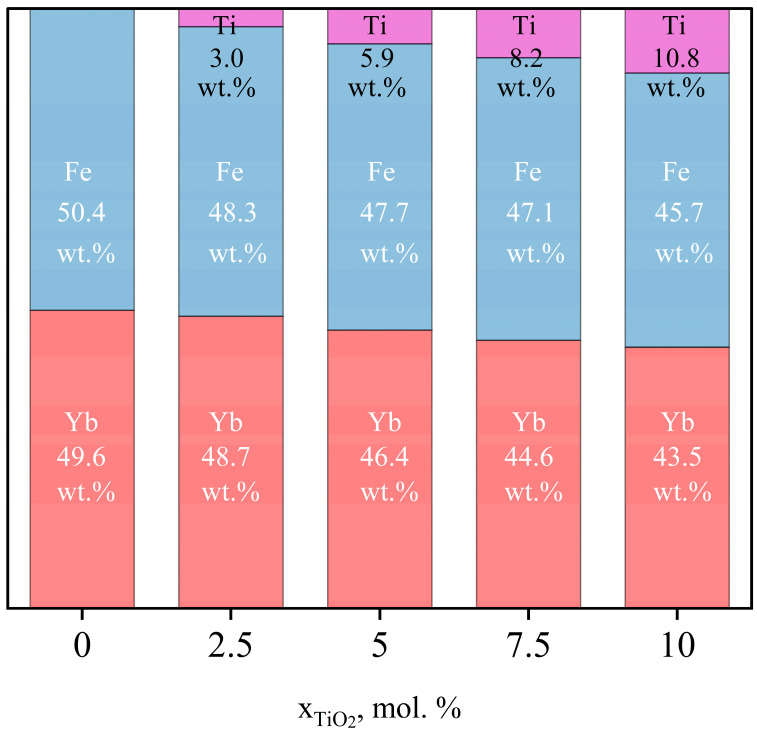
The elemental composition of samples concerning the main elements (in wt.%).

**Figure 2 materials-15-08273-f002:**
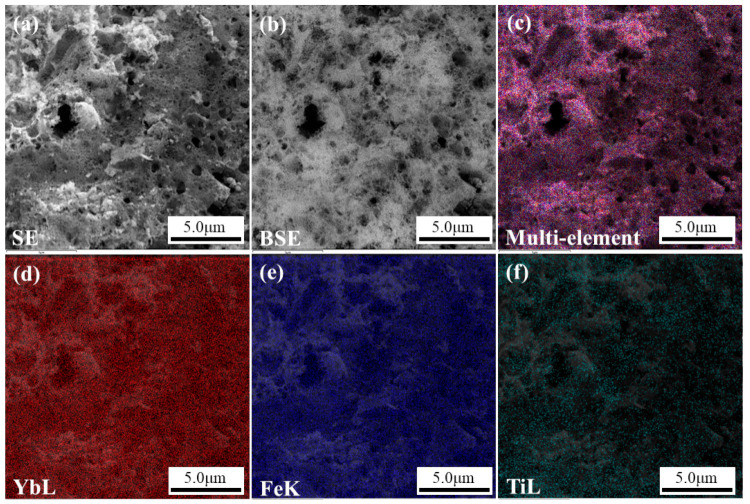
SEM images obtained on SE (**a**) and BSE (**b**) detectors and EDXS multi-element (**c**) and single element (**d**–Yb, **e**–Fe, **f**–Ti) mapping of a sample containing 5 mol.% of TiO_2_.

**Figure 3 materials-15-08273-f003:**
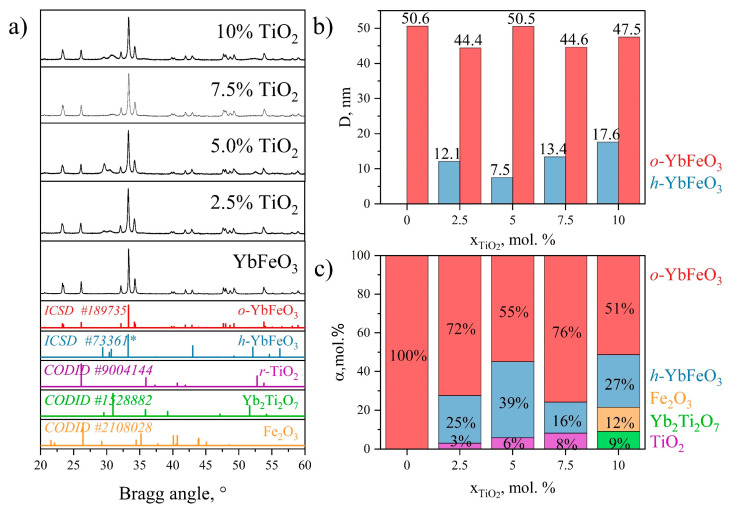
PXRD patterns of *o*-YbFeO_3_/*h*-YbFeO_3_/*r*-TiO_2_ (**a**), average crystallite sizes (D, nm) (**b**) and molar fraction (α, mol.%) (**c**) versus TiO_2_ content (mol.%).

**Figure 4 materials-15-08273-f004:**
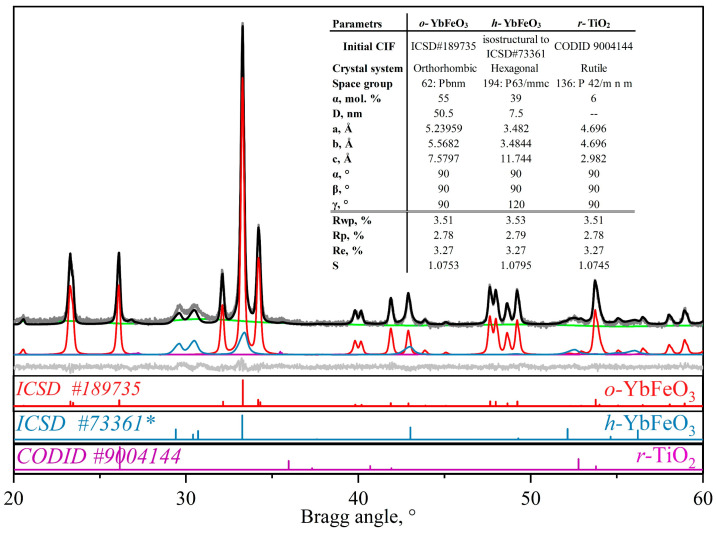
Rietveld refinement results for the X-ray diffraction of *o*-YbFeO_3_/*h*-YbFeO_3_/*r*-TiO_2_- 5%.

**Figure 5 materials-15-08273-f005:**
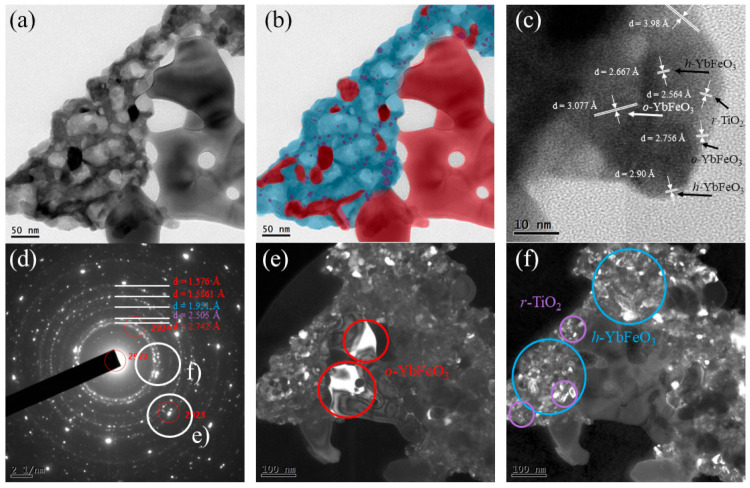
TEM images of the *h*-YbFeO_3_/*o*-YbFeO_3_/TiO_2_–5% sample: (**a**) an overview of the *o*-YbFeO_3_/*h*-YbFeO_3_/TiO_2_ nanocomposite, (**b**) color designation of phases: red—*o*-YbFeO_3_, blue—*h*-YbFeO_3_, purple—*r*-TiO_2_, (**c**) the *h*-YbFeO_3_ to *o*-YbFeO_3_ and to *r*-TiO_2_ heterojunction, (**d**) selected area transmission electron diffraction (SAED) pattern and dark-field TEM of *o*-YbFeO_3_ (**e**) and *h*-YbFeO_3_, *r*-TiO_2_ (**f**) nanocrystals.

**Figure 6 materials-15-08273-f006:**
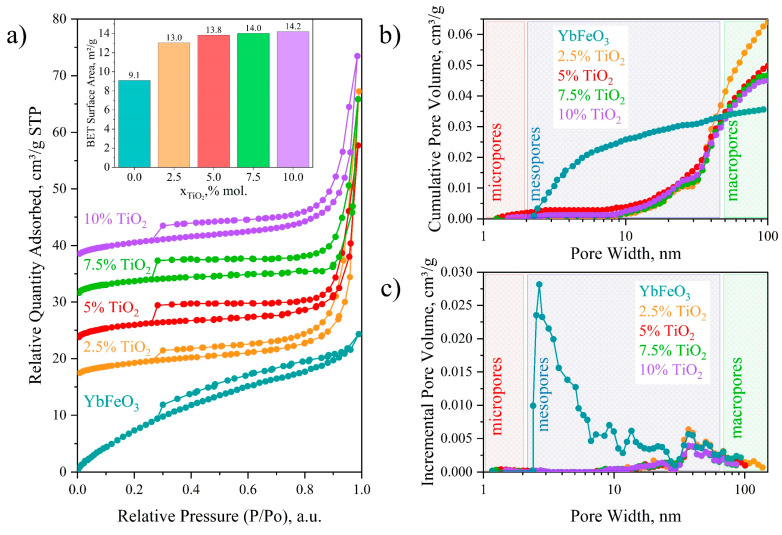
Low-temperature N_2_ sorption–desorption isotherms (**a**), cumulative (**b**), and incremental (**c**) pore volume distribution of *h*-YbFeO_3_/*o*-YbFeO_3_/TiO_2_ nanocomposites.

**Figure 7 materials-15-08273-f007:**
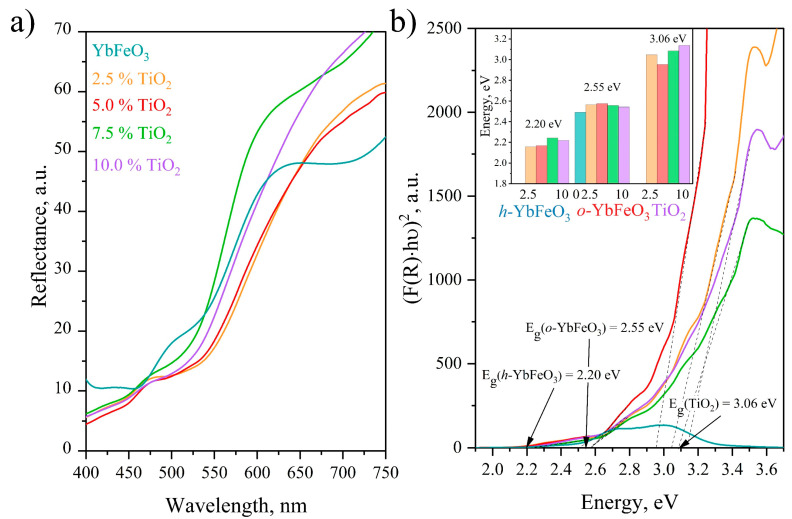
Diffuse reflectance spectra (**a**), Tauc plots (**b**), and band gap energy values (inset in b) of the *o*-YbFeO_3_/*h*-YbFeO_3_/TiO_2_ samples.

**Figure 8 materials-15-08273-f008:**
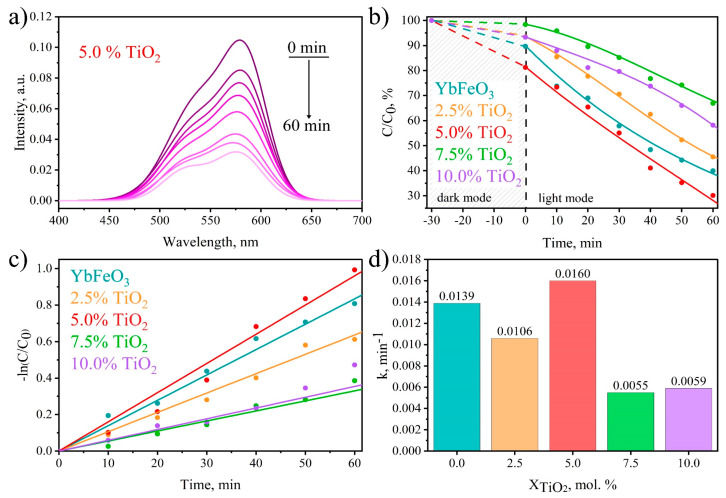
Photo-Fenton-like decomposition of methyl violet (MV) over *h*-YbFeO_3_/*o*-YbFeO_3_/TiO_2_–5% catalyst (**a**) kinetic curves (**b**) and logarithmic kinetic curves (**c**) and photodegradation reaction rate constant (**d**) versus TiO_2_ content.

**Figure 9 materials-15-08273-f009:**
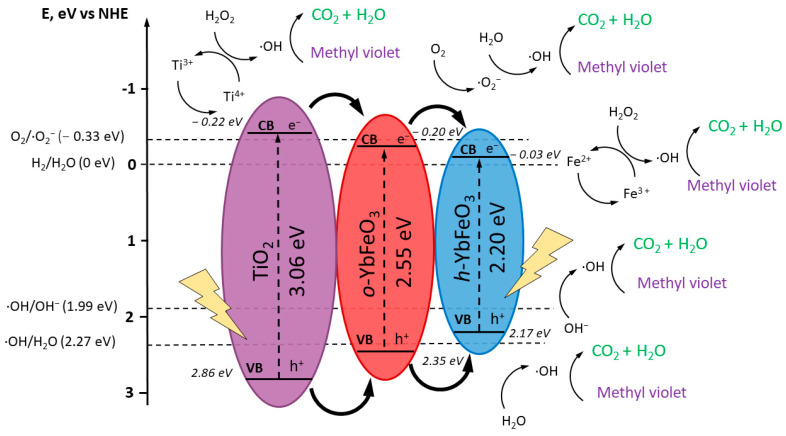
Schematic mechanism of photo-Fenton-like degradation of MV over *o*-YbFeO_3_/*h*-YbFeO_3_/TiO_2_ heterojunction nanocomposite.

**Table 1 materials-15-08273-t001:** Results of phase analysis by SAED.

Ring №	D-Spacing, Å	Ref. d-Spacing, Å	Hkl	Phase
1	2.742	2.780	0 2 0	*o*-YbFeO_3_
2	2.471	2.505	1 0 1	*r*-TiO_2_
3	1.950	1.951	0 0 6	*h*-YbFeO_3_
4	1.605	1.586	1 3 2	*o*-YbFeO_3_
5	1.373	1.376	3 2 2	*o*-YbFeO_3_

**Table 2 materials-15-08273-t002:** Parameters of the electronic structure of *o*-YbFeO_3_/*h*-YbFeO_3_/TiO_2_ nanocomposite.

	E_g_, ⋼B	χ, ⋼B;	E_VB_, ⋼B	E_CB_, ⋼B
***h*- ** **YbFeO_3_**	2.2	5.57	2.17	−0.03
***o*- ** **YbFeO_3_**	2.55	5.57	2.35	−0.20
**TiO_2_**	3.06	5.81	2.84	−0.22

## Data Availability

The data presented in this study are available on reasonable request from the corresponding author.

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
