# Peer review of "Effect of TiO2 Additives on the Stabilization of h-YbFeO3 and Promotion of Photo-Fenton Activity of o-YbFeO3/h-YbFeO3/r-TiO2 Nanocomposites"

_materials, 2022, doi:10.3390/ma15228273_

Round 1

Reviewer 1 Report

The manuscript required major revision:

1. The novelty of the work is missing in the introduction section. Explain it properly.

2. Improve language throughout the manuscript.

3. Similarity is too high (32%), make it to less than 20% and from a single paper, not more than 4%. Check the attached similarity report.

4. Improve the introduction based on the following reference on the nanocomposites and nanoparticles: (Radiation Effects and Defects in Solids Volume 174, Issue 5-6, Pages 480 - 4933 June 2019;  Results in Physics Open AccessVolume 13 June 2019 Article number 102264; Khan et al, 2 - Classification and properties of nanoparticles, 2022, Pages 15-54, ISBN 9780128242728)

Reviewer 2 Report

This paper reports on the synthesis and characterization of YbFeO3-TiO2 nanocomposites.
According to the authors' conclusions the samples have foamy morphology and highly porous structure, causing significant photocatalytic activity of the samples.
In addition the authors suggest that TiO2 phase stabilizes the h-YbFeO3 (critical for the photocatalytic activity). They also argue that these kind of nanocomposites can be used as promising photocatalysts for photo Fenton-like processes of oxidation of organic pollutants.

The paper contains a systematic well done experimental work, deserving a publication.

I have only one comment/suggestion.

I suggest the authors to perform a detailed Rietveld analysis of the powder x-ray diffraction data and to present Rietveld multi-phase plots in a figure, instead of / replacing Fig3.
The presentation with the help of ISCD cards is considered a bit old fashioned.

Reviewer 3 Report

Reviewer’s suggestions:

1.The reaction chemical formulas both in synthesis of YbFeO3-based photocatalysts and in synthesis of o-YbFeO3/h-YbFeO3/TiO2 nanocomposites should be presented in the “Materials and Methods” section of manuscript.

2.The influence of ferric ion in Fenton activity apart from the description of energy band gap mechanism should be also described how to produce the hydroxyl radicals at different spectrum in this study using Figure 8 in brief, like as the description of influence of TiO2 in hydroxyl radicals (·OH).

Reviewer 4 Report

What is the main question addressed by the research? Is it relevant and

interesting?

-The paper is about the Effect of TiO2 additives on the stabilization of h-YbFeO3 and 2

promotion of photo-Fenton activity of o-YbFeO3/h-YbFeO3/r- 3 TiO2 nanocomposites. The main problem was with the stability due to irreversible 10 transformation to a stable orthorhombic structure. The research is relevant and interesting as the authors report on the simple route to the stabilization of h-YbFeO3 nanocrystals by synthesis of multiphase nanocomposites with titania additives.

How original is the topic?

-The topic is new and has not been found elsewhere.

What does it add to the subject area compared with other published material?

-The nanocomposite can be used as promising pho- 325 tocatalysts for photo Fenton-like processes of oxidation of organic pollutants and other 326 advanced oxidation processes. Other publications have not addressed this in the past.

Is the paper well written?

-The paper is well written and easy to follow.

Is the text clear and easy to read?

-The text is clear and easy to read.

Are the conclusions consistent with the evidence and arguments presented?

-The conclusions are consistent with the results presented. The optimal amount of added TiO2 321 was 5%, which is due to the high values of the specific surface area and the proportion of 322 the hexagonal phase, as well as the processes of overlapping of the active centers of the 323 catalyst surface by TiO2 particles and an increase in charge recombination.

Do they address the main question

posed?

-Yes

Reviewer 5 Report

1. The conditions of experiments required to add in the Materials and Methods section, chapter 2.1.

2. Please, present in more detail the results achieved in this area of ​​research by other authors.

3.DRS-spectra, X-ray difraction are needed in more detail description.

Round 2

Reviewer 1 Report

Thanks for the revision. Manuscript is recommended for publication in current stage.

Author Response

Dear Reviewer, thank you for your attention to our work and helpful comments.